# Sleep Medication in Older Adults: Identifying the Need for Support by a Community Pharmacist

**DOI:** 10.3390/healthcare10010147

**Published:** 2022-01-13

**Authors:** Morgane Masse, Héloïse Henry, Elodie Cuvelier, Claire Pinçon, Margot Pavy, Audrey Beeuwsaert, Christine Barthélémy, Damien Cuny, Sophie Gautier, Nicolas Kambia, Jean-Marc Lefebvre, Daniel Mascaut, Fabrice Mitoumba, François Puisieux, Annie Standaert, Patrick Wierre, Jean-Baptiste Beuscart, Jean Roche, Bertrand Décaudin

**Affiliations:** 1Univ. Lille, CHU Lille, ULR 7365-GRITA—Groupe de Recherche sur les Formes Injectables et les Technologies Associées, F-59000 Lille, France; heloise.henry@univ-lille.fr (H.H.); elodie.cuvelier@univ-lille.fr (E.C.); christine.barthelemy@univ-lille.fr (C.B.); nicolas.kambia-kpakpaga@univ-lille.fr (N.K.); fabrice.mitoumba@univ-lille.fr (F.M.); bertrand.decaudin@univ-lille.fr (B.D.); 2Univ. Lille, CHU Lille, ULR2694-METRICS, Evaluation des Technologies de Santé et des Pratiques Médicales, F-59000 Lille, France; claire.pincon@univ-lille.fr (C.P.); Jeanbaptiste.BEUSCART@CHRU-LILLE.FR (J.-B.B.); 3Faculté de Pharmacie, Université de Lille, F-59000 Lille, France; margot.pavy@gmail.com (M.P.); audrey.beeuwsaert@gmail.com (A.B.); 4Univ. Lille, ULR 4515-LGCgE, Laboratoire de Génie Civil et Géo-Environnement, F-59000 Lille, France; damien.cuny@univ-lille.fr; 5Inserm, UMR-S1172, Center for Pharmacovigilance, CHU de Lille, University Lille, F-59037 Lille, France; sophie.gautier@chru-lille.fr; 6Department of General Practice, University of Lille, F-50045 Lille, France; docteurjml@icloud.com; 7Association des Conseillers et des Pharmaciens Agréés Maîtres de Stage du Nord—Pas-de-Calais, 21 Rue du Pont-Neuf, F-59000 Lille, France; daniel.mascaut@univ-lille.fr (D.M.); pfwierre@aol.com (P.W.); 8Hôpital Gériatrique Les Bateliers, Pôle de Gérontologie, CHRU de Lille, F-59000 Lille, France; Francois.PUISIEUX@CHRU-LILLE.FR; 9Univ. Lille, Inserm, CHU Lille, U1286-INFINITE—Institute for Translational Research in Inflammation, F-59000 Lille, France; annie.standaert@univ-lille.fr; 10CHU de Lille, Unité de Psychogériatrie, Pôle de Gérontologie, F-59037 Lille, France; Jean.ROCHE@chu-lille.fr

**Keywords:** sleep disorders, benzodiazepines, sedative-hypnotics, community pharmacy, older adults, sleep patterns

## Abstract

Many older adults take benzodiazepines and sedative-hypnotics for the treatment of sleep disorders. With a view to considering the possible discontinuation of hypnotics, the objectives of the present study were to describe bedtime habits and sleep patterns in older adults and to identify the sleep medications taken. An expert group developed a structured interview guide for assessing the patients’ bedtime habits, sleep patterns, and medications. During an internship in a community pharmacy, 103 sixth-year pharmacy students conducted around 10 interviews each with older adults (aged 65 or over) complaining of sleep disorders and taking at least one of the following medications: benzodiazepines, benzodiazepine derivatives (“Z-drugs”), antihistamines, and melatonin. A prospective, observational study was carried out from 4 January to 30 June 2016. The pharmacy students performed 960 interviews (with 330 men and 630 women; mean ± standard deviation age: 75.1 ± 8.8). The most commonly taken hypnotics were the Z-drugs zolpidem (*n* = 465, 48%) and zopiclone (*n* = 259, 27%). The vast majority of patients (*n* = 768, 80%) had only ever taken a single hypnotic medication. The median [interquartile range] prescription duration was 120 (48–180) months. About 75% (*n* = 696) of the patients had at least 1 poor sleep habit, and over 41% (*n* = 374) had 2 or more poor sleep habits. A total of 742 of the patients (77%) reported getting up at night—mainly due to nycturia (*n* = 481, 51%). Further, 330 of the patients (35%) stated that they were keen to discontinue their medication, of which 96 (29%) authorized the pharmacist to contact their family physician and discuss discontinuation. In France, pharmacy students and supervising community pharmacists can identify problems related to sleep disorders by asking simple questions about the patient’s sleep patterns. Together with family physicians, community pharmacists can encourage patients to discuss their hypnotic medications.

## 1. Introduction

Physiological sleep processes change as people age. Older adults have a longer sleep onset latency and more nocturnal awakenings. Overall, sleep in older adults is characterized by an earlier sleep phase, a reduction in the proportion of deep sleep, and the appearance of a short mid-afternoon nap [1]. Furthermore, the prevalence of sleep disorders (restless legs syndrome, parasomnia, sleep apnea, etc.) [2] and comorbid conditions that reduce sleep quality and quality of life increase with age; these difficulties are due (as least in part) to changes in circadian rhythms and lower melatonin levels [3,4].

Benzodiazepines and benzodiazepine derivatives (“Z-drugs”) target GABA receptors/pathways and are the most frequently prescribed psychotropic medications for sleep disorders [4]. The prevalence of benzodiazepine and sedative-hypnotic (BSH) use increases with the patient’s age [5]. In cases of insomnia, treatment with a BSH can reduce the sleep onset latency by a few minutes and increase the sleep period by an hour [6]. However, the broad prescription of BSHs is usually inappropriate: the prescription duration is too long, and patients may overdose by taking medication several times during the night [7]. BSHs have harmful long-term effects, such as deregulation of the sleep architecture and thus a further increase in sleep disorders. This is why BSH prescriptions are limited to 4 weeks: beyond that time, the harm outweighs the benefits [8]. Moreover, BSHs can have other harmful effects: daytime drowsiness, memory problems [9], and a nearly two-fold increase in the risk of falls [10]. Moreover, the polypharmacy often observed in older adults [11] increases the likelihood of drug interactions, adverse drug reactions (ADRs) [12,13,14], unplanned hospitalization, and death [15,16]. Z-drugs have been linked to an increased risk of falls and certain central nervous system (confusion, dizziness, daytime sleepiness, etc.) [10,17].

Given the high prevalence of inappropriate BSH prescriptions and the unfavorable associated risk-benefit ratio, prompt discontinuation of these sleep medications should be a health priority for older adults [18]. Discontinuation can be encouraged and facilitated by various interventions, including patient education, dose reduction, medication replacement, and psychological support [18,19]. The recent literature data suggest that pharmacist involvement can boost the deprescription of inappropriate medications [20,21,22,23].

However, psychological and pharmacological dependency on BSHs means that complete discontinuation is particularly challenging. Knowledge of the patient’s bedtime and sleep patterns, medications taken for sleep disorders and behavioral information can also facilitate discontinuation [24]. However, there are very few literature data on the patient’s bedtime and sleep patterns in older patients.

At the Lille Faculty of Pharmacy (Lille, France), the final-year internship always includes work related to patient care pathways [25,26]. In 2016, this work concerned sleep disorders in older adults.

The objectives of the present study were to describe bedtime and sleep patterns in older adults and to identify the sleep medications taken, notably with a view to prompting the patient to discuss the modification or discontinuation of hypnotic treatments with his/her family physician.

## 2. Materials and Methods

### 2.1. Study Design

This prospective, observational study was carried out between 4 January and 30 June 2016, in community pharmacies in the Nord—Pas-de-Calais region of France. One hundred and three sixth-year pharmacy student interns and their supervising community pharmacists reviewed medical records and dispensing histories; the selected patients were then invited to attend an interview. Each student was asked to interview about 10 older adults (aged 65 and over), complaining sleep disorders and taking at least one of medications: (i) benzodiazepines (i.e., estazolam, loprazolam, lormetazepam, nitrazepam, and temazepam), (ii) Z-drugs (zolpidem and zopiclone), (iii) antihistamines (i.e., hydroxyzine, alimemazine, doxylamine, and promethazine), and (iv) melatonin. Some of these hypnotics are not available outside of France. Patients treated for depression, psychosis or bipolar disorder were not included. The students invited the patients to be interviewed (with the agreement of the supervising community pharmacist) at home or at the pharmacy.

### 2.2. Ethical Aspects

The study was non-interventional; in line with French legislation, formal approval by an independent ethics committee was not therefore required. Prior to the interview, all the interviewees were given information about the study’s procedures and objectives and had the right to object to processing of their personal data. The interviewees also had the right to access, modify or remove their data on request at any time after the interview. The study was registered with the French National Data Protection Commission (*Commission nationale de l’informatique et des libertés* (Paris, France)) by the University of Lille’s (Lille, France) data protection officer.

### 2.3. Study Preparation

An expert working group (composed of 14 health professionals: clinical pharmacists, community pharmacists, pharmacologists, family physicians, geriatricians, and psychiatrists) drafted a number of documents required for standardized collection of the study data: (i) guidance on interviewing people and collecting information, (ii) an interview guide, and (iii) a letter for the interviewees’ family physicians.

The study preparation comprised (i) on-campus learning for the students and their supervising community pharmacists and (ii) online learning and tools. The students and their supervising community pharmacists were then trained in the use of the various documents. The interviewers also received training on sleep in older adults, sleep changes with age, medications for treating sleep disorders, ADRs, drug–drug interactions, and main contraindications for BSHs.

### 2.4. Interviews and Data Collection

The pharmacy students presented the project to the community pharmacy’s staff, detected potential interviewees, informed the family physician by mail or by phone, and (with the agreement of the patient and the supervising community pharmacist) made an appointment on behalf of the patient. Convenience sampling was used. The student prepared the interview using the patient’s dispensing records and conducted the interview by following the guide.

The data collection grid is provided as Appendix A. It included sections on the patient’s characteristics, medical history, number of prescription medications taken, and self-medication (if applicable). Particular attention was paid to the number of hypnotics taken, the presence/absence of overlaps between prescriptions of different medications, or recent changes in dosage. We collected information on the patient’s lifestyle, sleep (regularity of bedtime habits, sleep onset time, and the number and causes of nocturnal awakenings), the time before bedtime (presence of noise, light or electronic device use, and the bedroom temperature), sleep medications (duration of treatment, initiation in hospital or not, recent dose changes, self-medication, etc.). Data were missing for certain items. The students also asked the patient whether he/she had already considered discontinuing their medication, if they were ready to discontinue, and whether the pharmacist could contact the family physician to discuss the interview. After the interview, students updated the pharmacy’s records with regard to changes in hypnotic medications (dosage changes, medication switches, etc.).

The patient’s dependency on benzodiazepines was assessed on the Cognitive Scale for Benzodiazepine Attachment (*Echelle Cognitive d’Attachement aux Benzodiazépines*, ECAB) and was performed only among patients on benzodiazepines [27,28]. The ECAB questionnaire consists of 10 items scored as 1 or 0. The total score for the questionnaire is obtained by adding up the points from each item. A score of 6 or more corresponds to benzodiazepine dependency.

### 2.5. Statistical Analysis

Quantitative variables were expressed as mean ± standard deviation (SD) or median [interquartile range (IQR)], as appropriate. Qualitative variables were expressed as the frequency (percentage).

Logistic regression analysis was used to identify independent predictors of a wish to withdrawal hypnotic treatment. For each continuous variable, the log-linearity assumption was assessed with a visual inspection of the scatterplot between the empirical logits and the covariate, and was tested by comparing a model with the continuous covariate to a model including a quadratic component with an F-test for nested models. The multivariate model was built by including all predictors, using manual backward selection to reduce the model, by maximising the c-statistics. The final model was assessed with Hosmer–Lemeshow goodness of fit test. The two-sided Type I error rate was set at alpha = 0.05. All analyses were conducted with SAS software (version 9.4, SAS Institute Inc., Cary, NC, USA).

## 3. Results

### 3.1. Characteristics of the Study Population

The pharmacy students performed 960 patient interviews (with 330 men and 630 women; sex ratio: 0.52; mean ± standard deviation age: 75.1 ± 8.8). Almost half of the patients lived alone (*n* = 452, 47%). The most prevalent previous or concomitant conditions reported by the patients were cardiovascular disease (*n* = 723, 75%), dyslipidemia (*n* = 408, 43%), urinary disorders (*n* = 106, 11%), a fall in the previous 6 months (*n* = 192, 20%), and respiratory insufficiency (*n* = 75, 8%).

The median [interquartile] number of prescription medications taken daily was 6 [5,6,7,8,9]. Self-medication was reported by 15% of the patients (*n* = 145).

Nearly one-third of the patients (*n* = 330, 35%) stated that they had been willing at some point in the past to consider discontinuing medication. A total of 706 patients (74%) were not ready to try discontinuing and only 26% (*n* = 245) patients were willing at the time of the study. Only 10% (*n* = 94) of the patients gave their consent for the pharmacy staff to contact their family physician.

### 3.2. Bedtime Habits and Sleep Patterns

About 75% patients (*n* = 696) had at least one poor sleep habit, and more than 41% (*n* = 374) have two or more poor sleep habits (Table 1). Half of the patients (*n* = 533, 56%) had difficulty falling asleep, although 72% (*n* = 694) fell asleep within an hour (Figure 1). The mean ± SD sleep period was 8.5 ± 2.1 h. Almost three quarters of the patients woke up at least once at night (*n* = 742, 77%) (Table 2); this was mainly due to nycturia (*n* = 481, 51%). A total of 62% of the patients (*n* = 582) considered that they felt refreshed on wakening, and 35% patients (*n* = 334) had memory problems and/or difficulty concentrating.

### 3.3. Medications for Sleep Disorders

The vast majority of the patients (*n* = 768, 81%) had taken only one type of hypnotic medication in their lifetime; only 15% of the patients (*n* = 144) had tested two, and 4% (*n* = 36) had tested three or more. The most commonly taken hypnotics were benzodiazepine-derivatives (Z-drugs): 48% (*n* = 465) and 27% (*n* = 259) of the patients had taken used zolpidem and zopiclone, respectively (Table 3).

About 15% of the patients (*n* = 143) had started taking hypnotic medication during a hospital stay. The majority of patients took their hypnotic medication at around 10 pm (*n* = 365, 41%) (Figure 2). The vast majority of patients (*n* = 559, 63%) took their hypnotic medication at bedtime. The ECAB was scored for 176 of the 184 patients taking benzodiazepines; 51% (*n* = 89) had an ECAB score of 6 or more and so were considered to be benzodiazepine-dependent.

### 3.4. Factors Predictive of a Potential Wish to withdrawal Hypnotic Treatment

The multivariate analysis showed that young patients, a decrease in dosage in the last 6 months, a change in dosage by the patient himself and the patient’s management of the treatment were positive factors for the patient to wish to withdrawal treatment (Figure 3).

## 4. Discussion

The present study described older adults’ patterns and habits with regard to sleep and hypnotic medications. The study was based on 960 interviews. More than half reported having difficulty falling asleep, despite taking hypnotic medication. However, three quarters of the patients reported falling asleep within an hour. Our study also highlighted the elevated prevalence of poor habits and patterns before going to sleep. The vast majority of patients had tested only one hypnotic medication. At the end of the interview with the pharmacy student, nearly one-third of the patients (*n* = 330, 35%) stated that they had been willing to potentially discontinue medication at some point in the past, 245 (26%) of the 330 were willing to potentially discontinue medication at the time of the study, but only 94 agreed that the family physician could be contacted about possible deprescribing.

In older adults, sleep is characterized by frequent nocturnal awakenings, a reduction in the amount of deep sleep, early sleep onset, and early awakening [2]. The circadian mechanism becomes less efficient, and the overnight sleep period is often less than 6.0 to 7.5 h. In our study, however, the stated sleep period was higher; the mean ± SD value was 8.5 ± 2.1. Nearly 25% of the interviewees reported that they took more than 1 h to fall asleep. This fairly long time might be due (at least in part) to lifestyle patterns that do not promote sleep.

Indeed, many interviewees did not comply with good bedtime habits known to have a significant benefit for sleep (i.e., not using a screen in the bedroom, having a bedroom temperature below 19 °C (66.2 F°), etc.) [29,30]. Our results showed that about 75% of the patients had at least one poor sleep habit. A few reminders from the pharmacist about a healthy lifestyle might improve older adults’ sleep. A US study showed that telephone counseling on nutrition, physical activity, and sleep by a pharmacist can improve short-term physical and mental scores in the Duke Health Profile [31]. Moreover, the vast majority of our interviewees woke up during the night—mainly due to nycturia. Even though about half of interviewees woke up at night because of nycturia, only one in ten of the interviewees reported urinary disorders. This point could be addressed by the pharmacist through counselling on habits that might reduce the number of awakenings and nycturia (e.g., limiting late fluid intake, changing the administration time for diuretics and other medications that can cause nycturia, etc.) [32].

Although the patients were dissatisfied with their sleep, they appeared to be very committed to the medication they were taking: 8 in 10 of the patients had only tested a single hypnotic. The median time since the first prescription was 120 months, which is excessive and did not comply with current guidelines. In order to limit the side effects of hypnotics, the prescription period in France is limited to 28 days for zolpidem, nitrazepam, lormetazepam, zopiclone, estrazolam and loprazolam and to 12 weeks for hydroxyzine. Zolpidem was the most commonly used hypnotic. However, zolpidem has been classified as a narcotic since 2016; in order to limit the risk of abuse and misuse, the drug is subject to a controlled prescribing procedure [33]. This change led to a decrease in zolpidem prescriptions and a corresponding increase in zopiclone prescriptions. In France, it has been established that the two medications do not have the same effect: zolpidem is used to induce sleep, whereas zopiclone is used to maintain it [34]. The long-observed time since treatment initiation and the low number of patients with a recent dose change (126 of out 947, 13%) also reflects a lack of reassessment of a medication with many side effects. Half of the patients in our study were considered to be benzodiazepine-dependent, according to the ECAB score [27].

Melatonin and doxylamine are not related to benzodiazepines and so the prescription durations are not limited. Nevertheless, the two drugs should be used for short periods only. Sleep quality can be improved by changes in habits and patterns and not necessarily by medications. Doxylamine is present in many sleep medications but is classified as potentially inappropriate for older adults [18]. Although this medication is available over the counter, its use must be supervised by pharmacists [35].

One of our study’s most important findings was that about 15% of the hypnotic medications had been initiated during a hospital stay. Hospitalization can disturb sleep in older adults, due to environmental stimuli (such as noise or light exposure) and health problems [36]. However, the prescription for hypnotic medication was typically renewed after discharge from hospital, with no reassessment. This finding emphasizes the importance of re-evaluating medications on discharge for patients who were not previously being treated with hypnotics.

Another notable study finding concerned the patient’s willingness to discontinue the hypnotic medication. This is sometimes difficult for patients to envisage: only one third reported having already considered discontinuing their medication, and less than 25% patients felt ready to discontinue. Support and education are essential for successful discontinuation [23,37]. Martin et al. [23] showed that over-65 patients being followed up with an educational intervention were significantly more likely to discontinue benzodiazepine use than those without this follow-up. Cooperation between the family physician and the pharmacist is therefore essential for helping patients to discontinue BSHs [38]. Several studies have shown that a structured, personalized intervention helps to discontinue long-term benzodiazepine use [39,40]. However, only a small proportion of our interviewees authorized the community pharmacist to contact the family physician; this highlighting the moderate level of commitment to discontinuing sleep medications. It would be interesting to perform an interventional study assessing the number of patients who actually discontinued their medication following the interview with the pharmacist. Lastly, our results support the development of pharmacist interventions based on patient-centered inter-branch approaches. These approaches must take into account the patient’s behaviors and representations and must evolve from traditional “counselling” into actual motivational interventions [38]. Indeed, motivating patients is known to be an important aspect of deprescription strategies [41]. These strategies involve discussions about deprescription among members of the healthcare team; this team must include pharmacists and physicians because BSHs are prescription-only, controlled medications.

### Limitations

Our study has several limitations. Firstly, the participants were selected by pharmacy students during an internship with a community pharmacist; hence, the participants were recruited in a non-systematic way. Secondly, the interview was conducted (using a data collection grid) by pharmacy students, who do not have the experience of a senior pharmacist. Thirdly, the 6-month study duration was relatively short and prevented us from following up the patients. Fourthly, patients suffering from depression, psychosis or bipolar disorder were not included in our study, even though some were taking hypnotics. Fifthly, the study data were collected 5 years ago, and the patients’ sleep problems may have changed since then. Lastly, the study was performed in a single region of France, and so our results might be specific for the region’s cultural setting and/or the French healthcare system—especially since BSHs are widely prescribed in France [5].

## 5. Conclusions

Our results show that pharmacy students and supervising community pharmacists are well placed to identify problems related to patients’ sleep disorders. Special attention should be given to the patient’s lifestyle. Another key point concerns the reassessment of hypnotic medications, particularly upon discharge from hospital; medications were rarely reassessed, and the medication use often failed to comply with the summary of product characteristics. Community pharmacists could collaborate with family physicians to facilitate the discontinuation of hypnotic medications and increase the patient’s commitment to change. A discussion with patients about their sleep patterns, nycturia and fluid intake. Greater awareness of the ADRs associated with hypnotics might help to motivate the patients in this respect.

## Figures and Tables

**Figure 1 healthcare-10-00147-f001:**
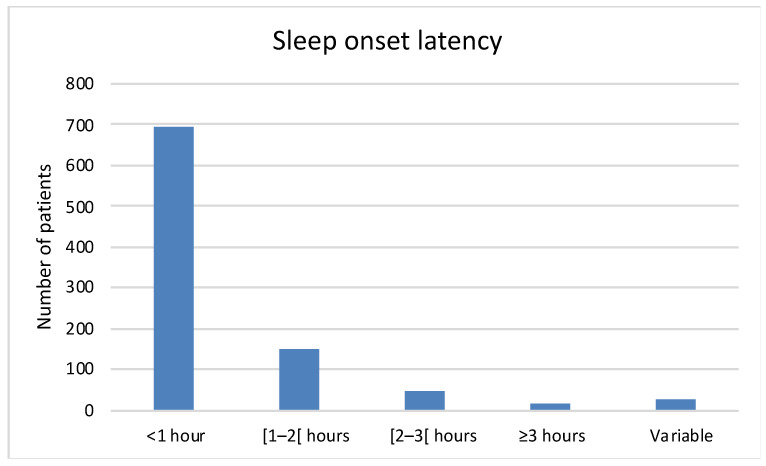
Sleep onset latency.

**Figure 2 healthcare-10-00147-f002:**
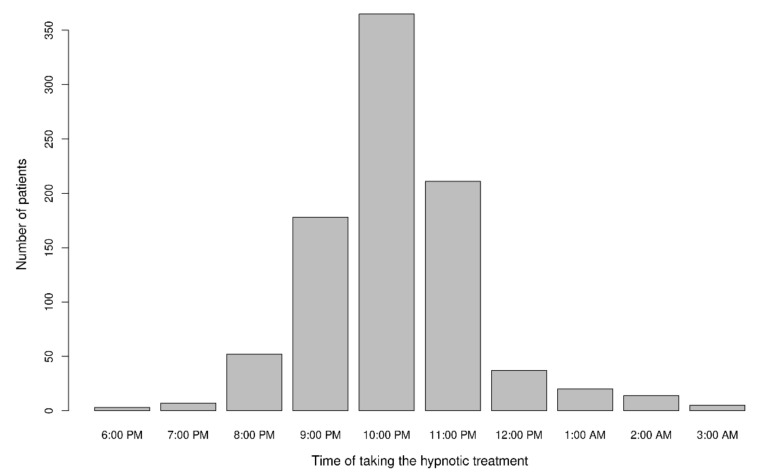
Time at which the hypnotic medication was taken (five patients took the hypnotic medication at different times in the evening/night). Data were missing for 63 of the 960 patients.

**Figure 3 healthcare-10-00147-f003:**
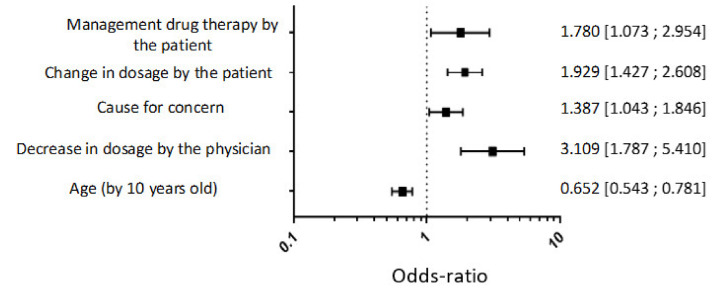
Multivariate logistic regression analysis to identify factors of a wish to withdrawal hypnotic treatment.

**Table 1 healthcare-10-00147-t001:** Bedtime habits and sleep patterns.

	Whole Population (*n* = 960)	Patients Were Not Ready to Try Discontinuing (*n* = 706)	Patients Who Were Ready to Try Discontinuing (*n* = 245)
Bedtime habits			
Facilitators			
Ritual before bedtime	694 (73%) ^1^	510 (71%)	184 (75%)
Barriers			
Large evening meal	128 (13%) ^2^	69 (10%)	59 (25%)
Alcohol consumption	268 (28%) ^1^	159 (22%)	109 (44%)
Tea/coffee consumption	123 (13%) ^1^	73 (10%)	50 (20%)
Disturbed by noise or light	194 (20%) ^3^	135 (19%)	59 (25%)
Screen in the bedroom	407 (42%) ^4^	300 (42%)	107 (44%)
Temperature >19 °C (66.2 F°)	216 (23%) ^5^	166 (23%)	50 (20%)
Sleep patterns			
Facilitators			
Regular bedtime	766 (80%) ^6^	576 (75%)	190 (25%)
Barriers			
Difficulty falling asleep	533 (56%) ^7^	396 (55%)	137 (56%)
Nocturnal awakening	742 (78%) ^8^	543 (76%)	199 (81%)

Data missing: ^1^ 11 (1%), ^2^ 9 (1%), ^3^ 6 (1%), ^4^ 4 (<1%), ^5^ 29 (3%), ^6^ 5 (1%), ^7^ 6 (1%), ^8^ 7 (1%).

**Table 2 healthcare-10-00147-t002:** Cause of nocturnal awakening. Data were missing for 9 of the 960 patients.

	Whole Sample (*n* = 960)	Patients Were Not Ready to Try Discontinuing (*n* = 706)	Patients Who Were Ready to Try Discontinuing (*n* = 245)
Nycturia	479 (50%)	342 (48%)	137 (56%)
Stress/anxiety	120 (13%)	86 (12%)	34 (14%)
Pain	95 (10%)	68 (10%)	27 (11%)
Nightmares	66 (7%)	40 (6%)	26 (11%)
Noise outside	65 (7%)	44 (6%)	21 (9%)
Spouse	57 (6%)	36 (5%)	21 (9%)
Snoring	46 (5%)	26 (4%)	20 (9%)
Noise in the house	32 (3%)	18 (3%)	14 (6%)
Respiratory problems, nocturnal cough	29 (3%)	21 (3%)	8 (3%)
Animals/pets	22 (2%)	17 (2%)	5 (2%)
Hot flushes	21 (2%)	11 (1%)	10 (4%)
Gastroesophageal reflux	21 (2%)	8 (1%)	13 (5%)

**Table 3 healthcare-10-00147-t003:** Hypnotic medications.

	Whole Sample (*n* = 960)	Patients Were Not Ready to Try Discontinuing (*n* = 706)	Patients Who Were Ready to Try Discontinuing (*n* = 245)
Current hypnotic medications [*n* (%)] ^1^			
Zolpidem	461 (48%)	354 (50%)	107 (44%)
Zopiclone	254 (27%)	183 (26%)	71 (29%)
Lormetazepam	105 (11%)	72 (10%)	33 (13%)
Loprazolam	43 (4%)	30 (4%)	13 (5%)
Estrazolam	25 (3%)	20 (3%)	5 (2%)
Hydroxyzine	21 (2%)	14 (2%)	7 (3%)
Doxylamine	16 (2%)	14 (2%)	2 (<1%)
Nitrazepam	9 (1%)	7 (1%)	2 (<1%)
Melatonin	7 (<1%)	5 (<1%)	1 (<1%)
Alimemazine	5 (<1%)	5 (<1%)	0 (0)
Temazepam	2 (<1%)	0 (0%)	2 (<1)
Promethazine	1 (<1%)	0 (0%)	1 (<1)
Dosage in compliance with the SPC [*n* (%)] ^2^	729 (78%)	543 (77%)	186 (76)
Duration of treatment (months) [median [interquartile range]] ^3^	120 (48–180)	108 (48–240)	120 (36–168)
Dosage change by the physician in the last 6 months [*n* (%)] ^4^	126 (13%)	79 (11%)	47 (19%)
Dosage change by the patient at some point [*n* (%)] ^5^	310 (33%)	187 (26%)	123 (50%)
Takes at least one more dose during the night [*n* (%)] ^6^	135 (14%)	94 (13%)	41 (17%)
Management of hypnotic medications by the patient him/herself [*n* (%)] ^7^	845 (88%)	619 (88%)	226 (92%)
ECAB score ^8^ ≥ 6	89 (51%)	63 (9%)	26 (11%)

Data missing: ^1^ 11 (1%), ^2^ 15 (2%), ^3^ 69 (7%), ^4^ 13 (1%), ^5^ 10 (1%), ^6^ 7 (1%), ^7^ 6 (1%), ^8^ 9 (5%); Abbreviations: SPC, summary of product characteristics; ECAB, Cognitive Scale for Benzodiazepine Attachment.

## Data Availability

Not applicable.

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
