# Peer review of "Sleep Medication in Older Adults: Identifying the Need for Support by a Community Pharmacist"

_healthcare, 2022, doi:10.3390/healthcare10010147_

Round 1

Author Response

We are extremely grateful for your interest in our manuscript and the reviewers’ very constructive comments. Please find enclosed a revised version of the manuscript entitled: "Sleep medication in older adults: identifying the need for support by a community pharmacist".

In line with the reviewers’ comments, we have improved the presentation and analysis of data and the contents in general. Point-by-point responses to the reviewers are given below.

Reviewer reports:

Reviewer 1

Thank you for the opportunity to review the manuscript, “Identifying sleep medication in older adults: need for support by a community pharmacists” for Healthcare.Inappropriate use of sleep medications is a significant problem among older adults. The large number of older adults that you were able to interview was impressive. In order to strengthen your manuscript, I recommend the following changes:

We thank the reviewer for his/her comment.

Abstract

  • Comment

Sentence 2 seems out of place. Consider removing.

Response

We have deleted this sentence.

  • Comment

Sentence 3 is a fragment.

Response

We have changed this sentence to: “With a view to considering the possible discontinuation of hypnotics, the objectives of the present study were to describe bedtime habits and sleep patterns in older adults and to identify the sleep medications taken.” (abstract – paragraph 1)

  • Comment

Consider putting the information in sequential order (i.e. introduce the interview guide and then talk about the pharmacy students)

Response

As suggested by the reviewer, we have changed the order of the sentences. We now start with the interview guide and follow it with the details on the pharmacy students: “An expert group developed the structured interview guide to assess the patients’ bedtime habits, sleep patterns, and medications. During an internship in a community pharmacy, 103 sixth-year pharmacy students conducted around 10 interviews each with older adults (aged 65 or over) complaining of sleep disorders and taking at least one of the following medications: benzodiazepines, benzodiazepine derivatives (“Z-drugs”), antihistamines, and melatonin.”

  • Comment

Giventhetitlefocusesonsleepmedications,considerprovidingtheresultsaboutthemedicationfirstfollowedbythebehaviors.

Response

As suggested by the reviewer, we have changed the order of the sentences: “A prospective, observational study was carried out from January 4th to June 30th, 2016. The pharmacy students performed 960 interviews (with 330 men and 630 women; mean ± standard deviation age: 75.1 ± 8.8). The most commonly taken hypnotics were the Z-drugs zolpidem (n=465, 48%) and zopiclone (n=259, 27%). The vast majority of patients (n=768, 80%) had only ever taken a single hypnotic medication. The median [interquartile range] prescription duration was 120 [48–180] months. About 75% (n=696) of the patients had at least 1 poor sleep habit, and over 41% (n=374) had 2 or more poor sleep habits. 742 of the patients (77%) reported getting up at night - mainly due to nycturia (n=481, 51%). 330 of the patients (35%) stated that they were keen to discontinue their medication, and 96 (10%) authorized the pharmacist to contact their family physician and discuss discontinuation”.

  • Comment

Considerchanging the sentence to “330 of the patients (35%) stated that they were keen to discontinue their medications, of which 96 (29%) authorized …” [Also,Ihaveanothernotebelowaskingforclarificationaboutwhetherthisshouldbe330or245patients.]

Response

Thank you for your suggestion. The difference between 330 and 245 relates to the time scale of the reply to the question about possible discontinuation: willingness at some point in the past for the 330, and willingness at the moment of the study for the 245. We have modified the sentence, as follows:“330 of the patients (35%) stated that they were keen to discontinue their medication, of which 96 (29%) authorized the pharmacist to contact their family physician and discuss discontinuation.” (abstract – paragraph 3)

  • Comment

Recommend changing the second to last sentence to state, “Pharmacy students in the community setting can identify problems …” as that aligns with your methods.

Response

We have modified the sentence: “In France, pharmacy students and supervising pharmacists in community setting can identify problems related to sleep disorders by asking simple questions about the patient’s sleep patterns”. (abstract –paragraph 4)

Introduction

  • Comment

Recommend adjusting the sentence associated with reference 17 to state, “Given the highprevalenceofinappropriateBSHprescriptionsandtheunfavorableassociatedrisk-benefitratio,promptdiscontinuationofthesesleepmedicationsshouldbea health priority for older adults”.  

Response

Thank you for your suggestion, we have modified the sentence as recommended.(introduction –3 paragraph)

  • Comment

Thelastsentenceindicatesthatacommunitypharmacistisdoingthiswork.However,itseemsthatstudentpharmacistswereactuallycarryingoutthe interviews.

Response

We have modified the sentence as follows: “The objectives of the present study were to describe bedtime and sleep patterns in older adults and to identify the sleep medications taken, notably with a view to prompting the patient to discuss the modification or discontinuation of hypnotic treatments with his/her family physician.” (abstract –paragraph 4)

Materials and methods

  • Comment

Study preparation: consider including the total number of people in the working group

Response

We have modified the sentence as follows:“The working group was composed of 14 health professionals (clinical pharmacists, community pharmacists, pharmacologists, family physicians, geriatricians and psychiatrists)”. (Materials and methods - 2.3.Study preparation – paragraph 1)

  • Comment

Study preparation: how was training conducted? It would be helpful to include sufficient details so that others could replicate this program.

Response

The study preparation comprised (i) on-campus (face-to-face) learning for the students and their supervising community pharmacists and (ii) online learning and tools. We now specify this in the manuscript: “The study preparation was composed of blended learning: on-campus learning for the students and their community pharmacists supervising and online instruction”. (Materials and methods - 2.3.Study preparation – paragraph 2)

  • Comment

It is unclear if you were only seeking to gather data from participants or if there was also an intervention (e.g., statements encouraging participants to talk with their prescriber about stopping the medication).

Response

The study in itself did not seek to develop systematic or standardized interventions related to sleep medications. However, giving advice is part of the pharmacist’s role. If problems were identified during the interview, the student could give some advice (e.g. sleep management, and encouragement to discuss possible treatment with the interviewee’s family physician).

Results

  • Comment

Do you know where the remaining 53% of participants lived? It would be very different if most of them lived with a spouse at home vs. an assisted living facility vs. facility providing higher amounts of care.

Response

All the patients included in our study were living at home. The remaining 53% lived at home but with someone (i.e. not alone).

  • Comment

Consider making a figure that shows the number of poor sleep habits per participant. While you have information about individual behaviors in table 1, it is not possible to see how they related to one another.

Response

We have modified Table 1 to make it more readable and have added the following sentence to the text: “About 75% patients (n=696) had at least 1 poor sleep habit, and more than 41% (n=374) have 2 or more poor sleep habits (Table 1)”. (Results - 3.2.Bedtime habits and sleep patterns – paragraph 1)

Table 1.Bedtime habits and sleep patterns.

Patients were not ready (n=715)

Patients who were ready to try discontinuing again (n=245)

Whole population (n=960)

Bedtime habits

Facilitators

Ritual before bedtime

510 (71%)

184 (75%)

694 (73%)1

Barriers

Large evening meal

69 (10%)

59 (25%)

128 (13%)2

Alcohol consumption

159 (22%)

109 (44%)

268 (28%)1

Tea/coffee consumption

73 (10%)

50 (20%)

123 (13%)1

Disturbed by noise or light

135 (19%)

59 (25%)

194 (20%)3

Screen in the bedroom

300 (42%)

107 (44%)

407 (42%)4

Temperature >19°C (66.2 F°)

166 (23%)

50 (20%)

216 (23%)5

Sleep patterns

Facilitators

Regular bedtime

576 (75%)

190 (25%)

766 (80%)6

Barriers

Difficulty falling asleep

396 (55%)

137 (56%)

533 (56%)7

Nocturnal awakening

543 (76%)

199 (81%)

742 (78%)8

Data missing:  111 (1%), 29 (1%), 36 (1%), 44 (<1%), 529 (3%), 65 (1%), 76 (1%), 87 (1%)

  • Comment

Potential wish to discontinue medication already considered discontinuing their medication. At the time of the interview, only 26% (n=245) patients were ready to try discontinuation again, of which 38% (n=94) gave their consent for the pharmacy staff to contact their family physician. However, this represents only 10% of the total. Do you have any data about the 94 patients who gave consent to contact the prescriber? Was the prescriber receptive? Did they lower the dose or frequency or stop the medication? If not, consider discussing this in the discussion or limitations section.

Response

We did not analyze the data for these 94 patients and therefore did not know if they actually decreased or stopped their treatments.

In fact, this study was primarily learning-based, rather than interventional. As the course only lasted 6 months, it was not possible to extend the follow-up period.

We have specified this aspect in the limitations paragraph: “Thirdly, the study duration of 6 months was relatively short, not allowing for patient follow-up”. (Discussion – Limitations)

Discussion

  • Comment

It is confusing if 245 or 330 patients were actually interested in deprescribing their sleep medication at this time.

Response

We thank the reviewer for this comment. Indeed, 330 patients were initially open to stopping medication but only 245 patients were ultimately open to stopping it. We have modified the sentence:“Nearly one-third of the patients (n=330, 35%) stated that they had been willing at some point in the past to consider discontinuing medication but only 26% (n=245) patients were willing at the time of the study.”(Discussion – paragraph 1)

Conclusion

  • Comment

The conclusion should clarify that the interviews were conducted by student pharmacists.

 Response

We have modified the sentence as follows:“Our results show that pharmacy students and supervising community pharmacists are well placed to identify problems related to patients’ sleep disorders.” (conclusions)

Table 1

  • Comment

- Recommendaccountingformissingdataviaafootnoteinsteadofasanadditionalcolumn

- Recommend decreasing amount of text in the description (e.g., “alcohol consumption with the meal” could be “alcohol consumption” “tea/coffee consumption in the evening” could either be “tea or coffee consumption” or “caffeine consumption”

- Rightnow,thereisamixoffacilitators/positiveexperiencesandbarriers/negativeexperiencestosleep.Recommendeitherdividingthemtwocolumnsinthesametable(seeexamplebelow)or create two separate tables. Alternatively, you could change them all to negative items (e.g. “did not report ritual”)

- Thesleeponsetlatencyandcauseofnocturnalawakeningisdifficulttoreadinthistable.Considerwhethertheinformationneedstobehereorcouldbemovedtoeitherthetextoraseparatetableorfigure.

- Recommend adding a footnote to specifying which items were “select all that apply” (e.g., cause of nocturnal awakening?)          

Facilitators             Numberofparticipants(percent)  Barriers     Numberofparticipants(percent)

B

efore bed

Ritual or routine

694 (73)

Large evening meal

128 (13)

Alcohol consumption

268 (28)

Caffeine consumption

123 (13)

Response

We have complied with the journal’s instructions to authors and have modified Table 1 to make it more readable.

As suggested:

- We havedecreased the amount of text in the description

- We haveseparatedfacilitators/positiveexperiences on one handandbarriers/negativeexperiencestosleep on the other.

- We have added (i) a new table on the cause of nocturnal awakening and (ii) a figure on sleep onset latency

Patients were not ready (n=715)

Patients who were ready to try discontinuing again (n=245)

Whole population (n=960)

Bedtime habits

Facilitators

Ritual before bedtime

510 (71%)

184 (75%)

694 (73%)1

Barriers

Large evening meal

69 (10%)

59 (25%)

128 (13%)2

Alcohol consumption

159 (22%)

109 (44%)

268 (28%)1

Tea/coffee consumption

73 (10%)

50 (20%)

123 (13%)1

Disturbed by noise or light

135 (19%)

59 (25%)

194 (20%)3

Screen in the bedroom

300 (42%)

107 (44%)

407 (42%)4

Temperature >19°C (66.2 F°)

166 (23%)

50 (20%)

216 (23%)5

Sleep patterns

Facilitators

Regular bedtime

576 (75%)

190 (25%)

766 (80%)6

Barriers

Difficulty falling asleep

396 (55%)

137 (56%)

533 (56%)7

Nocturnal awakening

543 (76%)

199 (81%)

742 (78%)8

Data missing:  111 (1%), 29 (1%), 36 (1%), 44 (<1%), 529 (3%), 65 (1%), 76 (1%), 87 (1%)

Table 2

  • Comment

Recommendorderingmedicationsfrommosttoleastfrequent

Response

We have ordered the medications from most to least frequent:

Whole sample

(n = 960)

Current hypnotic medications [n(%)]1

Zolpidem

465 (48%)

Zopiclone

259 (27%)

Lormetazepam

105 (11%)

Loprazolam

42 (4%)

Estrazolam

25 (3%)

Hydroxyzine

21 (2%)

Doxylamine

16 (2%)

Nitrazepam

10 (1%)

Melatonin

6 (<1%)

Alimemazine

5 (<1%)

Temazepam

2 (<1%)

Promethazine

1 (<1%)

Dosage in compliance with the SPC* [n(%)]2

735 (78%)

Duration of treatment (months) [median [interquartile range]]3

120 [48 - 180]

Dosage change by the physician in the last 6 months [n(%)]4

126 (13%)

Dosage change by the patient at some point [n(%)]5

313 (33%)

Takes at least one more dose during the night [n(%)]6

137 (14%)

Management of hypnotic medications by the patient him/herself [n(%)]7

850 (89%)

ECAB** score8 ³ 6

89 (51%)

Data missing: 13 (0%), 212 (1%), 369 (7%), 413 (1%), 510 (1%), 67 (1%), 76 (1%),  89 (5%)

* SPC: summary of product characteristics

** ECAB: échelle cognitive d'attachement aux benzodiazépines- Cognitive Scale for Benzodiazepine Attachment

  • Comment

Recommendaddingafootnotetospecifyifmultipleproductscouldbeselected

Response

Only one drug was taken by the patient.

  • Comment

Recommendremovingmissingdatacolumnandmovingthisinformationtoafootnote

Response

We have complied with the journal’s instructions to authors.

Figure

  • Comment

Thetitleofthefigureisunclear – “Figure 63 of the 960 participants”

Response

The title of the figure is: Time at which the hypnotic medication was taken (five patients took the hypnotic medication at different times in the evening or night). Data were missing for 63 of the 960 patients.”

Reviewer 2 Report

Masse et al devised a new survey that community pharmacists can use to measure the behavioral sleep habits and tendencies of older adults taking pharmacotherapy to treat insomnia. The authors stated their intent of reporting behavioral information was that it could be used by clinicians to better “facilitate discontinuation” of sedative hypnotics (introduction paragraph 4). However, I believe their analysis falls a bit short of fully exploring this premise. The authors simply reported their aggregate findings of their survey results, but did not explore the differences between those ready to deprescribe compared to those who were not ready. Delineating differences is very important because it can help generate further hypotheses related to proper patient targeting for hypnotic deprescribing interventions. For instance, if we knew that patients with shorter sleep onset latencies were more amendable to deprescribing interventions, we would know that this factor is important to assess the feasibility of future deprescribing interventions.   

Overall, the paper was well written and is a worthwhile addition to the literature simply because of the potential practical usefulness of the survey. However, the descriptive analysis that was conducted needs more expansion as discussed above. This is the major revision. I did include several other minor comments that will improve the clarity of the manuscript.

  1. Abstract: 3rd sentence "To describe...": This is a rough transition. Should be rewritten as "Our objective was to describe..."
  2. Introduction: Second paragraph. The authors introduce and seem to define “BSH” as benzos and z-drugs. Several of these references (6,8,9) only refer to benzodiazepines. Please substantiate the sentences with references that are also specific to z-drugs. Moreover, the authors capture antihistamines and melatonin in their analysis yet do not introduce these drugs as potentially problematic. So, the authors should make a short reference to these agents as well and their potential risks.
  3. Introduction: As is, the objective is awkwardly written. Perhaps, “the objective was to report the bedtime behaviors and sleep patterns in older adults who were pharmacologically treated for insomnia.”
  4. Methods:  section 2.1 "Patients treated for depression, psychosis, etc...": Two points: 1. How were these conditions identified? Medication profile serving as a proxy? Or identified during patient review? Or….? 2. By excluding these conditions, you are cutting out a substantial number of patients who have insomnia since these conditions are after comorbid. Please address in the limitations section. 
  5. Methods: section 2.1: Please note that several of these drugs are not available in other countries.
  6. Methods: section 2.2: please report registration number.
  7. Methods: section 2.4: how were patients actually recruited? at the point of presenting a new RX for a sedative hypnotic? Selecting the first ## of patients in a queried report for sedative hypnotics prescriptions?
  8. Methods section 2.4: The link to ECAB fails, so please provide a new reference. Also, is this a validated tool? Can you explain more details regarding this tool? Also, was the ECAB administered to pts on z drugs and anticholinergics? If so, is it valid for those classes? Or, was this just administered to those on benzos?
  9. results: 3.1: Elaborate on the definition of urinary disorders (UI?), cardiovascular disease, neuro disease? 
  10. Results: 3.2: first sentence is not in table 1: what defines a poor sleep habit?
  11. Results: 3.2 please remove "only" from the last sentence: “only 62%...”
  12. Table 1: nycturia: If possible, I would like some analysis/ investigation (or explanation) of other drug-causes (loops before bed?)/medical causes (fluid intake before bed?). It’s really remarkable that 10% had a urinary disorder yet 50% had symptoms of nocturia... 
  13. Table 2: Sort hypnotics by prevalence please.
  14. Table 2: since there are only 4 other hypnotics in the footnote, place this into the table with the n(%). the table isn't that large to begin with.
  15. Table 2: I think it’s worthwhile mentioning somewhere that temazepam is a non-CYP drug that preferred in elderly. However, it wasn’t very prevalent at all. There could be some discussion around better prescribing patterns specific to older adults.
  16. Results: section 3.3: "only 15%..." eliminate the only.
  17. Results: section 3.4: As stated above, please compare the characteristics of the people who were ready to try discontinuing again (n=245) compared to those who were not ready (n=715). What factors were significantly differed between groups? Consider demographic and clinical factors in addition to the sleep behaviors derived from the survey.
  18. Discussion: paragraph 1: How do we know they were “dissatisfied”? Most didn't want to stop! no data to back this judgment up. Please re-write or revise accordingly.
  19. Discussion: paragraph 1: “…330 patient said…” I think this is an error. Shouldn’t this be 245?
  20. Discussion: page 7, paragraph 2: “zopiclone is used to maintain…” Is this specific to France? In the US, it’s indicated for both.
  21. Discussion: page 7, paragraph 3: “potentially inappropriate for older adults.” Need a reference here (e.g., 2019 Beers criteria).
  22. Discussion: page 7, paragraph 5: “Another strength…” This isn’t really a strength. It’s an observation. I feel as though you’re actually missing an important strength: your survey. It could be very helpful for community pharmacists who consult patients with insomnia.

Author Response

Reviewer 2

Masse et al devised a new survey that community pharmacists can use to measure the behavioral sleep habits and tendencies of older adults taking pharmacotherapy to treat insomnia.

  • Comment

The authors stated their intent of reporting behavioral information was that it could be used by clinicians to better “facilitate discontinuation” of sedative hypnotics (introduction paragraph 4). However, I believe their analysis falls a bit short of fully exploring this premise. The authors simply reported their aggregate findings of their survey results, but did not explore the differences between those ready to deprescribe compared to those who were not ready. Delineating differences is very important because it can help generate further hypotheses related to proper patient targeting for hypnotic deprescribing interventions. For instance, if we knew that patients with shorter sleep onset latencies were more amendable to deprescribing interventions, we would know that this factor is important to assess the feasibility of future deprescribing interventions.  

Overall, the paper was well written and is a worthwhile addition to the literature simply because of the potential practical usefulness of the survey. However, the descriptive analysis that was conducted needs more expansion as discussed above. This is the major revision. I did include several other minor comments that will improve the clarity of the manuscript.

Response

In fact, we studied factors associated with wishing to stop treatment prior to our first submission of the manuscript. We performed univariate and multivariate regressions (see the figure summarizing the results below). None of the factors were strongly associated with patients' wishes.

However, these factors did not appear to be relevant for clinical pharmacy practice, and we believe that the pharmacist’s dedication of more time to the patient would be more patient-friendly.

We have chosen not to present these results so as not to weigh down the "statistical data" and "supplementary data" sections. However, if the reviewer insists and the editor agrees, we could include the results of the univariate and multivariate regressions.

“The results were: “The multivariate analysis showed that young patients, a decrease in dosage in the last 6 months, a change in dosage by the patient himself and the patient's management of the treatment were positive factors for the patient to wish to withdrawal treatment.

Figure: Multivariate logistic regression analysis to identify factors of a wish to withdrawal hypnotic treatment.”

Abstract

  • Comment

3rd sentence "To describe...": This is a rough transition. Should be rewritten as "Our objective was to describe..."

Response

We are agree with the reviewer and have modified the sentence: “With a view to considering the possible discontinuation of hypnotics, the objectives of the present study were to describe bedtime habits and sleep patterns in older adults and to identify the sleep medications taken.” (abstract - paragraph 1)

Introduction

  • Comment

Second paragraph. The authors introduce and seem to define “BSH” as benzos and z-drugs. Several of these references (6,8,9) only refer to benzodiazepines. Please substantiate the sentences with references that are also specific to z-drugs.

Response

The French High Authority for Health’s official guide (reference 9) refers to benzodiazepines andZ-drugs (imidazopyridine derivatives like zolpidem and cyclopyrrolone derivatives like zopiclone). We have added the following sentence: “Z-drugs have been linked to an increased risk of falls and certain central nervous system (confusion, dizziness, daytime sleepiness, etc.(references: Zolpidem: Efficacy and Side Effects for Insomnia and Z-drugs and risk for falls and fractures in older adults-a systematic review and meta-analysis)”. (introduction - paragraph 2)

  • Comment

Moreover, the authors capture antihistamines and melatonin in their analysis yet do not introduce these drugs as potentially problematic. So, the authors should make a short reference to these agents as well and their potential risks.

Response

As this was a learning-focused study, the goal was to identify patients with sleep problems; hence, all hypnotic drugs were included, regardless of their side effects.

  • Comment

As is, the objective is awkwardly written. Perhaps, “the objective was to report the bedtime behaviors and sleep patterns in older adults who were pharmacologically treated for insomnia.”

Response

We thank you for this suggestion and have modified the objectives accordingly: “The objectives of the present study were to describe bedtime and sleep patterns in older adults and to identify the sleep medications taken, notably with a view to prompting the patient to discuss the modification or discontinuation of hypnotic treatments with his/her family physician.” (introduction - paragraph 5)

Methods 

  • Comment

Section 2.1 "Patients treated for depression, psychosis, etc...": Two points: 1. How were these conditions identified? Medication profile serving as a proxy? Or identified during patient review? Or….? 2. By excluding these conditions, you are cutting out a substantial number of patients who have insomnia since these conditions are after comorbid. Please address in the limitations section.

Response

After the investigators had reviewed all the patients’ medication and dispensing histories, selected patients were invited to attend an interview. We have added these sentences: “One hundred and threesixth-year pharmacy student interns and their supervising community pharmacists reviewed medical records and dispensing histories; the selected patients were then invited to attend an interview”. (methods -2.1. Study design paragraph 1)

Although this exclusion parameter restricted the scope of the study, our goal was to study a targeted intervention in a well-defined population. We have added this point to the limitations paragraph: “Fourthly, patients suffering from depression, psychosis or bipolar disorder were not included in our study, even though some were taking hypnotics”. (discussion – limits)

  • Comment

Section 2.1: Please note that several of these drugs are not available in other countries.

Response

We specified that: “This prospective, observational study was carried out from January 4thto June 30th, 2016, in community pharmacies in the Nord - Pas-de-Calais region of France”. We have added: “Some of these hypnotics are not available outside of France”.(methods -2.1. Study design paragraph 1)

  • Comment

Section 2.2: please report registration number.

Response

We have added: “The study was registered with the French National Data Protection Commission (Commission nationale de l'informatique et des libertés (Paris, France)) by the University of Lille’s (Lille, France) data protection officer.”(methods -2.2. Ethical aspects - paragraph 1)

  • Comment

Section 2.4: how were patients actually recruited? at the point of presenting a new RX for a sedative hypnotic? Selecting the first ## of patients in a queried report for sedative hypnotics prescriptions?

Response

Patients were recruited when the prescription containing a hypnotic drug was dispensed, in agreement with the supervising community pharmacists.

We have added the following sentence: “One hundred and threesixth-year pharmacy student interns and their supervising community pharmacists reviewed medical records and dispensing histories; the selected patients were then invited to attend an interview”.(methods -2.1. Study design paragraph 1)

  • Comment

Section 2.4: The link to ECAB fails, so please provide a new reference. Also, is this a validated tool? Can you explain more details regarding this tool? Also, was the ECAB administered to pts on z drugs and anticholinergics? If so, is it valid for those classes? Or, was this just administered to those on benzos?

Response

We apologize for that error and have provided the correct link: https://has-sante.fr/upload/docs/application/pdf/2009-10/9_ecab_scale_vf.pdf. This is the Benzodiazepine Cognitive Attachment Scale, used to identify benzodiazepine-dependent patients. This scale is not valid for other drug class (Z drugs, anticholinergics, melatonin, etc.).

Results

  • Comment

3.1: Elaborate on the definition of urinary disorders (UI?), cardiovascular disease, neuro disease?

Response

The pharmacy students classified the following urinary complaints as “urinary disorders”: urinary incontinence, pollakiuria, and urinary urgency.

The cardiovascular diseases included all such disease, from high blood pressure to chronic heart failure.

Neurologic disease included stroke, Parkinson’s disease, etc

This classification was not standardized: the student classified the disease on the basis of his/her knowledge or asked the supervising pharmacist.

  • Comment

3.2: first sentence is not in table 1: what defines a poor sleep habit?

Response

We have modified Table 1 to make it more readable.

Patients were not ready (n=715)

Patients who were ready to try discontinuing again (n=245)

Whole population (n=960)

Bedtime habits

Facilitators

Ritual before bedtime

510 (71%)

184 (75%)

694 (73%)1

Barriers

Large evening meal

69 (10%)

59 (25%)

128 (13%)2

Alcohol consumption

159 (22%)

109 (44%)

268 (28%)1

Tea/coffee consumption

73 (10%)

50 (20%)

123 (13%)1

Disturbed by noise or light

135 (19%)

59 (25%)

194 (20%)3

Screen in the bedroom

300 (42%)

107 (44%)

407 (42%)4

Temperature >19°C (66.2 F°)

166 (23%)

50 (20%)

216 (23%)5

Sleep patterns

Facilitators

Regular bedtime

576 (75%)

190 (25%)

766 (80%)6

Barriers

Difficulty falling asleep

396 (55%)

137 (56%)

533 (56%)7

Nocturnal awakening

543 (76%)

199 (81%)

742 (78%)8

Data missing:  111 (1%), 29 (1%), 36 (1%), 44 (<1%), 529 (3%), 65 (1%), 76 (1%), 87 (1%)

  • Comment

3.2 please remove "only" from the last sentence: “only 62%...”

Response

We have removed the word “only”, and the sentence now reads: “62% of the patients (n=582) considered that they felt refreshed on wakening, and 35% patients (n=334) had memory problems and/or difficulty concentrating”.

  • Comment

Table 1: nycturia: If possible, I would like some analysis/ investigation (or explanation) of other drug-causes (loops before bed?)/medical causes (fluid intake before bed?). It’s really remarkable that 10% had a urinary disorder yet 50% had symptoms of nocturia...

Response

These results corresponded to statements by the patient. As the suggested by the reviewer, nycturia might have been drug-related or due to fluid intake. We did not collect data on evening diuretic use and so could not analyze this variable.

  • Comment

Table 2: Sort hypnotics by prevalence please.

Response

We have sorted the hypnotics by prevalence.

Whole sample (n = 960)

Missing data

Current hypnotic medications [n(%)]

3 (0%)

Zolpidem

465 (48%)

Zopiclone

259 (27%)

Lormetazepam

105 (11%)

Loprazolam

42 (4%)

Estrazolam

25 (3%)

Hydroxyzine

21 (2%)

Doxylamine

16 (2%)

Nitrazepam

10 (1%)

Melatonin

6 (<1%)

Alimemazine

5 (<1%)

Temazepam

2 (<1%)

Promethazine

1 (<1%)

  • Comment

Table 2: since there are only 4 other hypnotics in the footnote, place this into the table with the n(%). the table isn't that large to begin with.

Response

We have added the percentages for melatonin, alimemazine, temazepam, and promethazine in Table 2.

Whole sample (n = 960)

Current hypnotic medications [n(%)]1

Zolpidem

465 (48%)

Zopiclone

259 (27%)

Lormetazepam

105 (11%)

Loprazolam

42 (4%)

Estrazolam

25 (3%)

Doxylamine

16 (2%)

Hydroxyzine

21 (2%)

Nitrazepam

10 (1%)

Melatonin

6 (<1%)

Alimemazine

5 (<1%)

Temazepam

2 (<1%)

Promethazine

1 (<1%)

Data missing: 13 (0%)

  • Comment

Table 2: I think it’s worthwhile mentioning somewhere that temazepam is a non-CYP drug that preferred in elderly. However, it wasn’t very prevalent at all. There could be some discussion around better prescribing patterns specific to older adults.

Response

In our study, zolpidem and zopiclone were the most prescribed. In France, these hypnotics are very widely prescribed. The frequency depends on a country’s prescribing habits. We have noted in the limitations paragraph that our study was performed in a single region of France.

  • Comment

Section 3.3: "only 15%..." eliminate the only.

Response

We have modified this sentence: “About 15% of the patients…”

  • Comment

Section 3.4: As stated above, please compare the characteristics of the people who were ready to try discontinuing again (n=245) compared to those who were not ready (n=715). What factors were significantly differed between groups? Consider demographic and clinical factors in addition to the sleep behaviors derived from the survey.

Response

We have modified Table 1 so that it compares the characteristics of the people who were ready to try discontinuing again (n=245) with those of the people who were not (n=715).

Discussion

  • Comment

Paragraph 1: How do we know they were “dissatisfied”? Most didn't want to stop! no data to back this judgment up. Please re-write or revise accordingly.

Response

We have removed the sentence: “Overall, the interviewees were dissatisfied with their hypnotic medication”.

  • Comment

Paragraph 1: “…330 patient said…” I think this is an error. Shouldn’t this be 245?

Response

You are right: 330 patients had initially considered discontinuing their medication but only 245 patients were ready to try again. We have added: “At the end of the interview with the pharmacy student, nearly one-third of the patients (n=330, 35%) stated that they had been willing to potentially discontinue medication at some point in the past, 245 (26%) of the 330 were willing to potentially discontinue medication at the time of the study, but only 94 agreed that the family physician could be contacted about possible deprescribing.” (discussion – paragraph 1)

  • Comment

Page 7, paragraph 2: “zopiclone is used to maintain…” Is this specific to France? In the US, it’s indicated for both.

Response

Although zolpidem and zopiclone have the same indication in the marketing authorization for France, their respective prescriptions depends on the physician’s habits. We have modified the sentence as follows: “In France, it has been established that the two medications do not have the same effect: zolpidem is used to induce sleep, whereas zopiclone is used to maintain it”. (discussion – paragraph 3)

  • Comment

Page 7, paragraph 3: “potentially inappropriate for older adults.” Need a reference here (e.g., 2019 Beers criteria).

Response

We have added this reference.

  • Comment

Page 7, paragraph 5: “Another strength…” This isn’t really a strength. It’s an observation. I feel as though you’re actually missing an important strength: your survey. It could be very helpful for community pharmacists who consult patients with insomnia.

Response

We have modified this sentence: “Another noticeable point …” (discussion – paragraph 6)

Reviewer 3 Report

Comments on MDPI manuscript “Sleep medication in older adults: identifying the need for support by a community pharmacist”

I want to thank the journal editors and the authors for reading the above manuscript. This manuscript makes an interesting contribution to the scientific area of healthcare services provided by pharmacists.

Next, authors can find my comments for their perusal.

One general comment pertains to the possible disconnection between surveying patients and referring patients to other healthcare professionals. One thing is the pharmacist’s autonomy to screen for health and treatment issues; another is to solve those patients’ health problems effectively. Looking at the title, the manuscript should address the authors’ views on implementing such pharmacists’ screening services and the feasibility of networking with GPs.

The authors make the large sample size (960 interviewees) very clear throughout the manuscript. The authors never link this information with power calculations since non-probabilistic sampling was followed. Nevertheless, the figure’s repetition without further elaboration (e.g. in Discussion) suggest a potential or surrogate statistical representation, which is not the case. Please, make sure to avoid this underlying idea.

Additionally, data is five years old, and in between, the world witnessed a pandemic with clear implications on elders’ (and everyone) health. Authors should address this in the study limitations.

The manuscript Discussion mostly restates the study findings without elaboration (or speculation) by, e.g. comparison with previous studies. The authors mentioned a small number of studies (I have literature suggestions later in my comments); thus, the Discussion is overall poor in references, which should be improved.

Abstract.

The title mention ‘pharmacists’. The abstract intro suggests that anyone, from elders themselves to any formal or informal carer, could be involved in family doctors’ referral. This intro seems less coherent with the title.

Abstract sentences need more linkage. For instance, it is hard to understand the sentence “To describe bedtime and sleep patterns in older adults and to identify the sleep medications taken” as the study aim. Authors should add, “The primary study objectives were to describe … “ This applies to the other sections.

The interview guide used a grid. As such, the sentence should read, “An expert group developed the structured interview guide… “.

I am not aware of pharmacy practice in France, regulations, and so on. I guess the responsible pharmacist handled the contact with the family doctor. Thus, I find it strange that a licensed healthcare professional facing a Drug-Related Problem (DRP) need patient authorization to work with the prescriber. In my opinion, it is an ethical obligation. The authors follow a medical-centric perspective regarding healthcare provision, which conflicts with the title.

As well, the passage “Community pharmacists can identify problems related to sleep disorders by asking simple questions about the patient’s sleep patterns. Together with family physicians, community pharmacists can encourage patients to discuss of hypnotic treatment approach” are showing that the authors may be unaware of the long-established discipline of Clinical Pharmacy. The use and abuse of Z-drugs is a global issue, and pharmacists’ role and collaboration with prescribers have been studied for decades. Thus, I recommend framing the study into the local or regional context, i.e., mentioning France or the Lille area, since conclusions are not necessarly new for an international audience.

Introduction

1st paragraph. I would recommend avoiding three sentences without references, even if these belong to references 1 and 2.

4th paragraph. Authors wrote, “However, there are no literature data on these topics in older adults,” including “medications taken for sleep disorders.” Several countries run national surveys on sleeping issues, such as in the USA (National Sleep Foundation – Sleep in America Survey) or the European Sleep Research Society. At least one publication (van de Straat V, Bracke P. How well does Europe sleep? A cross-national study of sleep problems in European older adults. International journal of public health. 2015 Sep;60(6):643-50.) deals with the issue in Europe. The same authors have recently published a study regarding older adults and medication (van de Straat V, Buffel V, Bracke P. Medicalization of sleep problems in an ageing population: a longitudinal cross-national study of medication use for sleep problems in older European adults. Journal of ageing and health. 2018 Jun;30(5):816-38). Additionally, there are publications with pharmacists’ interventions in sleep disorders, such as Fuller JM, Wong KK, Krass I, Grunstein R, Saini B. Sleep disorders screening, sleep health awareness, and patient follow-up by community pharmacists in Australia. Patient education and counselling. 2011 Jun 1;83(3):325-35. I recommend the authors revisit the literature or stress the lack of publications when referring to their reality.

Two last paragraphs. Please clarify if the internship status is not an undergraduate position, i.e., previous to receiving full professional autonomy as a registered pharmacist.

Material and Methods

The sampling type should be mentioned earlier since this is a study design option.

The inclusion/exclusion criteria do not mention other conditions such as dementia (not infrequent in elders), which may hamper the quality of the data collected by an interview. There are quick screening tools for mental disability to participate in research studies (e.g. mini-mental state exam). Why were these not used?

In Study Preparation, I was expecting that students would receive written instructions on patient recruitment and selection. On the other hand, since data collection was made using a grid, I guess authors followed a structured interview, although nothing is mentioned regarding the validity and/or reliability of such guide (and grid). For instance, no piloting or pre-testing is mentioned. This issue is relevant to the high number of variables (some of qualitative nature) mentioned in the following subheading. In particular, asking older patients about stopping such a medication requires careful questioning and good interviewing skills. This is even more critical when the authors mention that interviewers received training, raising issues of standardization with 103 persons.

Students had the opportunity of visiting patients for home-based interviews. It is a well-known the possible recall and memory bias in elderly patients (sample mean age 75 years). Why did not the students ask for a ‘brown bag’ medication review (at home or the pharmacy), matching self-reported medications with the actual drugs being used? The authors mention that data was missing for some items, as well as discrepancies with pharmacies records.

Related to the guide validity and reliability, the authors cited the instrument used for assessing Z-drug dependency. However, it is missing the literature sources for items such as bedtime habits and sleep patterns.

Results

How long did the interviews take? Any concerns about the interview length with elders?

Regarding the sleep variables subheadings, I would recommend highlighting (e.g. with indentation and italic): Sleep onset latency, Cause of nocturnal awakening, Sleep period, etc. This aspect is somewhat better in Table 2, although it also needs a review for increasing legibility.

“…35% patients (n=334) had memory problems and/or difficulty concentrating.” How sure are the authors that these complaints result from sleeping issues and not another health issue? Was variables association tested?

Why are authors providing a graph (Figure 63) with medication times? Does this address any of the study objectives? How does this relates to discontinuation or the pharmacists’ work?

Discussion

The sentence “Most of the interviewees were active: they collected their own medications from the pharmacy and managed their own medications” belongs to the Results. It would be best to elaborate or speculate on the implications of this finding while responding to your study objectives. Only in the 3rd paragraph there is a reference to a US study.

The Discussion is scarce on contrasting the present findings with the existing evidence. Are the current results aligned or not with the existing body of knowledge? For instance, what are the implications of having 330 patients saying one thing and behaving oppositely? The actual willingness regarding Z-drugs discontinuation paves the way to move from restating results and bringing novel topics in pharmaceutical care, such as motivational interventions. Also, it allows for assumptions regarding the (poor) societal image of community pharmacists, (low) expected level of intervention and the degree of (in)visibility as healthcare professionals.

Motivational interventions may play an essential role in patients’ Z-drugs management, including behaviour change techniques (BCTs). Pharmacists (and physicians) should move from the traditional ‘counselling’ (i.e. an informative role) to actual motivational interventions. The authors mention the need for patient support and education as one of the most relevant study deliverables (last Discussion paragraph). In my opinion, this approach is somewhat outdated. For example, brief motivational interventions are much more efficient in reaching target behaviours, such as changing medication use patterns.

Knowing the gap in medication reassessment and benzo-dependence, authors could elaborate on pharmacists’ roles towards closely monitoring this DRP. What can pharmacists do to help elderly patients and their physicians with more rational and safer hypnotic medication use? Therefore, the authors should not forget the role of prescribers in this context. These medications are prescription-only and controlled drugs. Deprescribing is also needed, and pharmacists can contribute to reducing society over-medicalization.

Thank you.

Author Response

We are extremely grateful for your interest in our manuscript and the reviewers’ very constructive comments. Please find enclosed a revised version of the manuscript entitled: "Sleep medication in older adults: identifying the need for support by a community pharmacist".

In line with the reviewers’ comments, we have improved the presentation and analysis of data and the contents in general. Point-by-point responses to the reviewers are given below.

Reviewer reports:

Reviewer 3

I want to thank the journal editors and the authors for reading the above manuscript. This manuscript makes an interesting contribution to the scientific area of healthcare services provided by pharmacists.

Next, authors can find my comments for their perusal.

One general comment pertains to the possible disconnection between surveying patients and referring patients to other healthcare professionals. One thing is the pharmacist’s autonomy to screen for health and treatment issues; another is to solve those patients’ health problems effectively. Looking at the title, the manuscript should address the authors’ views on implementing such pharmacists’ screening services and the feasibility of networking with GPs.

  • Comment

The authors make the large sample size (960 interviewees) very clear throughout the manuscript. The authors never link this information with power calculations since non-probabilistic sampling was followed. Nevertheless, the figure’s repetition without further elaboration (e.g. in Discussion) suggest a potential or surrogate statistical representation, which is not the case. Please, make sure to avoid this underlying idea.

Response

In this study, the number of older people included was based on pragmatic considerations and not power calculations. Each of the 103 students was encouraged to conduct 10 interviews.

  • Comment

Additionally, data is five years old, and in between, the world witnessed a pandemic with clear implications on elders’ (and everyone) health. Authors should address this in the study limitations.

Response

We now mention in the limitations paragraph that the data were collected 5 years ago. The details of the sleep problems may have changed since then. We have added the following sentence: “Fifthly, the study data were collected 5 years ago, and the patients’ sleep problems may have changed since then.” (discussion – limitations)

  • Comment

The manuscript Discussion mostly restates the study findings without elaboration (or speculation) by, e.g. comparison with previous studies. The authors mentioned a small number of studies (I have literature suggestions later in my comments); thus, the Discussion is overall poor in references, which should be improved.

Response

We have specifically addressed these issues in the following comments.

Abstract.

  • Comment

The title mention ‘pharmacists’. The abstract intro suggests that anyone, from elders themselves to any formal or informal carer, could be involved in family doctors’ referral. This intro seems less coherent with the title.

Response

We have deleted the sentence: A person wishing to modify or discontinue his/her medication should be referred to his/her family physician. The pharmacist is an important player in the support of these patients.

Abstract sentences need more linkage.

  • Comment

For instance, it is hard to understand the sentence “To describe bedtime and sleep patterns in older adults and to identify the sleep medications taken” as the study aim. Authors should add, “The primary study objectives were to describe … “ This applies to the other sections.

Response

We have modified the sentence: “With a view to considering the possible discontinuation of hypnotics, the objectives of the present study were to describe bedtime habits and sleep patterns in older adults and to identify the sleep medications taken.” (Abstract)

  • Comment

The interview guide used a grid. As such, the sentence should read, “An expert group developed the structured interview guide… “.

Response

We thank you for this suggestion and have modified the sentence: “An expert group developed a structured interview guide for assessing the patients’ bedtime habits, sleep patterns, and medications.”(abstract – paragraph 1)

  • Comment

I am not aware of pharmacy practice in France, regulations, and so on. I guess the responsible pharmacist handled the contact with the family doctor. Thus, I find it strange that a licensed healthcare professional facing a Drug-Related Problem (DRP) need patient authorization to work with the prescriber. In my opinion, it is an ethical obligation. The authors follow a medical-centric perspective regarding healthcare provision, which conflicts with the title.

Response

In France, personal medical information can be shared among healthcare professionals but only if the patient agrees. Hence, in the context of the study, it was essential for the interviewer to check with the patient before phoning or otherwise containing the patient’s family physician. The patients had agreed to participate in the study on a voluntary basis, and the potential problem was identified by the interviewer and not by the patient. In France, most patients would not want their pharmacist to contact their family physician without their prior approval. This enables the patient to keep control of his/her own health care pathway. Overall, this possible contact with the family physician was  a way of promoting collaborative self-management, rather than a regulatory requirement.

  • Comment

As well, the passage “Community pharmacists can identify problems related to sleep disorders by asking simple questions about the patient’s sleep patterns. Together with family physicians, community pharmacists can encourage patients to discuss of hypnotic treatment approach” are showing that the authors may be unaware of the long-established discipline of Clinical Pharmacy. The use and abuse of Z-drugs is a global issue, and pharmacists’ role and collaboration with prescribers have been studied for decades. Thus, I recommend framing the study into the local or regional context, i.e., mentioning France or the Lille area, since conclusions are not necessarly new for an international audience.

Response

We have mentioned that the study was conducted in France.

Introduction

  • Comment

1st paragraph. I would recommend avoiding three sentences without references, even if these belong to references 1 and 2.

Response

We have added a reference for the second sentence of the paragraph (Dzierzewski, J.M.; Dautovich, N.; Ravyts, S. Sleep and Cognition in Older Adults. Sleep Med. Clin.2018, 13, 93–106, doi:10.1016/j.jsmc.2017.09.009).

  • Comment

4th paragraph. Authors wrote, “However, there are no literature data on these topics in older adults,” including “medications taken for sleep disorders.” Several countries run national surveys on sleeping issues, such as in the USA (National Sleep Foundation – Sleep in America Survey) or the European Sleep Research Society. At least one publication (van de Straat V, Bracke P. How well does Europe sleep? A cross-national study of sleep problems in European older adults. International journal of public health. 2015 Sep;60(6):643-50.) deals with the issue in Europe. The same authors have recently published a study regarding older adults and medication (van de Straat V, Buffel V, Bracke P. Medicalization of sleep problems in an ageing population: a longitudinal cross-national study of medication use for sleep problems in older European adults. Journal of ageing and health. 2018 Jun;30(5):816-38). Additionally, there are publications with pharmacists’ interventions in sleep disorders, such as Fuller JM, Wong KK, Krass I, Grunstein R, Saini B. Sleep disorders screening, sleep health awareness, and patient follow-up by community pharmacists in Australia. Patient education and counselling. 2011 Jun 1;83(3):325-35. I recommend the authors revisit the literature or stress the lack of publications when referring to their reality.

Response

Thank you for suggesting these references. We meant to say that there were very little data on the patients' sleep patterns, rather than on medications taken for their sleep disorders. We have modified the sentence as follows: “However, there are very few literature data on the patient’s bedtime and sleep patterns in older adults.” (Introduction – paragraph 4)

  • Comment

Two last paragraphs. Please clarify if the internship status is not an undergraduate position, i.e., previous to receiving full professional autonomy as a registered pharmacist.

Response

Internship is an undergraduate position, previous to receiving full professional autonomy as a registered pharmacist. We have added this reference: Olivier Bourdon et al.Pharmacy education in France, Am J Pharm Educ. 2008 Dec 15;72(6):132.  doi: 10.5688/aj7206132.

Material and Methods

  • Comment

The sampling type should be mentioned earlier since this is a study design option.

Response

We have added a sentence:“This prospective, observational study was carried out between January 4thand June 30th, 2016, in community pharmacies in the Nord - Pas-de-Calais region of France.” (Material and Methods - 2.1.Study design – paragraph 1)

  • Comment

The inclusion/exclusion criteria do not mention other conditions such as dementia (not infrequent in elders), which may hamper the quality of the data collected by an interview. There are quick screening tools for mental disability to participate in research studies (e.g. mini-mental state exam). Why were these not used?

Response

The evaluation of cognitive disorders was beyond the scope of the present study. We agree that this point would be interesting, since recent studies have shown interactions and associations between Alzheimer's disease or other dementias and sleep disorders. However, cognitive evaluations would have required other ethical approvals and would also have hampered the study. Consequently, the question of cognitive disorders could only be addressed through the patients’ own statements.

  • Comment

In Study Preparation, I was expecting that students would receive written instructions on patient recruitment and selection. On the other hand, since data collection was made using a grid, I guess authors followed a structured interview, although nothing is mentioned regarding the validity and/or reliability of such guide (and grid). For instance, no piloting or pre-testing is mentioned. This issue is relevant to the high number of variables (some of qualitative nature) mentioned in the following subheading. In particular, asking older patients about stopping such a medication requires careful questioning and good interviewing skills. This is even more critical when the authors mention that interviewers received training, raising issues of standardization with 103 persons.

Response

The grid was not validated in a preliminary phase. The study preparation comprised (i) on-campus (face-to-face) learning for the students and their supervising community pharmacists and (ii) online learning and tools. This is now mentioned in the manuscript: “The study preparation comprised (i) on-campus learning for the students and their supervising community pharmacists and (ii) online learning and tools”. (Material and Methods - 2.3.Study preparation– paragraph 2)

  • Comment

Students had the opportunity of visiting patients for home-based interviews. It is a well-known the possible recall and memory bias in elderly patients (sample mean age 75 years). Why did not the students ask for a ‘brown bag’ medication review (at home or the pharmacy), matching self-reported medications with the actual drugs being used? The authors mention that data was missing for some items, as well as discrepancies with pharmacies records.

Response

The students looked at the dispensing history and noted hypnotics taken for sleep problems and the dose level prescribed.

The students did not ask for a ‘brown bag’ medication review (at home or at the pharmacy) because we believe that the history of the drugs was sufficient. The students relied on the long-standing dispensing information in the pharmacist's computer system. Therefore, the students were aware of all the medications dispensed to the patient. The missing data concerned the patients’ sleep patterns, rather the medications taken.

  • Comment

Related to the guide validity and reliability, the authors cited the instrument used for assessing Z-drug dependency. However, it is missing the literature sources for items such as bedtime habits and sleep patterns.

Response

We have added a reference for the French health authority’s guide on sleep disorders and dependency (available athttps://www.has-sante.fr/jcms/r_1500930/fr/troubles-du-sommeil-stop-a-la-prescription-systematique-de-somniferes-chez-les-personnes-agees)

Results

  • Comment

How long did the interviews take? Any concerns about the interview length with elders?

Response

That aspect is very interesting but unfortunately is one that we did not study.

  • Comment

Regarding the sleep variables subheadings, I would recommend highlighting (e.g. with indentation and italic): Sleep onset latency, Cause of nocturnal awakening, Sleep period, etc. This aspect is somewhat better in Table 2, although it also needs a review for increasing legibility.

Response

We have modified Table 1 to make it more readable.

  • Comment

 “…35% patients (n=334) had memory problems and/or difficulty concentrating.” How sure are the authors that these complaints result from sleeping issues and not another health issue?

Response

We are not claiming that the memory problems and/or difficulty concentrating were related to hypnotics. The complaints were probably also related to cognitive impairment, bearing in mind that sleep disturbance may be a sign or consequence of underlying dementia. A detailed investigation of this aspect was beyond the scope of the study.

  • Comment

Why are authors providing a graph (Figure 1) with medication times? Does this address any of the study objectives? How does this relates to discontinuation or the pharmacists’ work?

Response

The figure shows the times at which medications were taken. Although most patients took their sleep medication around 10pm, some took it after midnight. Knowing the time of administration was important because the community pharmacist could advise the latter patients to take the medication earlier in the evening.

Discussion

  • Comment

The sentence “Most of the interviewees were active: they collected their own medications from the pharmacy and managed their own medications” belongs to the Results. It would be best to elaborate or speculate on the implications of this finding while responding to your study objectives. Only in the 3rd paragraph there is a reference to a US study.

Response

We have deleted the sentence: Most of the interviewees were active: they collected their own medications from the pharmacy and managed their own medications. However, many hadchronic diseases and some had geriatric syndromes; for example, 1 in 5 interviewees had fallen in the previous 6 months.

  • Comment

- The Discussion is scarce on contrasting the present findings with the existing evidence. Are the current results aligned or not with the existing body of knowledge? For instance, what are the implications of having 330 patients saying one thing and behaving oppositely? The actual willingness regarding Z-drugs discontinuation paves the way to move from restating results and bringing novel topics in pharmaceutical care, such as motivational interventions. Also, it allows for assumptions regarding the (poor) societal image of community pharmacists, (low) expected level of intervention and the degree of (in)visibility as healthcare professionals.

- Motivational interventions may play an essential role in patients’ Z-drugs management, including behaviour change techniques (BCTs). Pharmacists (and physicians) should move from the traditional ‘counselling’ (i.e. an informative role) to actual motivational interventions. The authors mention the need for patient support and education as one of the most relevant study deliverables (last Discussion paragraph). In my opinion, this approach is somewhat outdated. For example, brief motivational interventions are much more efficient in reaching target behaviours, such as changing medication use patterns.

- Knowing the gap in medication reassessment and benzo-dependence, authors could elaborate on pharmacists’ roles towards closely monitoring this DRP. What can pharmacists do to help elderly patients and their physicians with more rational and safer hypnotic medication use? Therefore, the authors should not forget the role of prescribers in this context. These medications are prescription-only and controlled drugs. Deprescribing is also needed, and pharmacists can contribute to reducing society over-medicalization

Response

We have added the following sentences at the end of the Discussion: “Lastly, our results support the development of pharmacist interventions based on patient-centered inter-branch approaches. These approaches must take into account the patient's behaviors and representations and must evolve from traditional “counselling” into actual motivational interventions [36]. Indeed, motivating patients is known to be an important aspect of deprescription strategies [39]. These strategies involve discussions about deprescription among members of the healthcare team; this team must include pharmacists and physicians because BSHs are prescription-only, controlled medications.”.

Round 2

Reviewer 2 Report

Overall most of my comments were addressed. There are still a few points that are unclear after your revision.

  1. I can appreciate that the authors do want want to convolute the message regarding the univariate and multivariate analyses that identify factors that may be associated with a higher likelihood of wanting to deprescribe. However, I still I think you should include these analyses in some capacity.  Perhaps the data can be placed in the appendix as supplemental tables to avoid - as you state - weighing your paper down with stats. Then, summarize the results as you did in your response to me in the results section. Also, it would be helpful to revise tables 2 and 3 similar your revised table 1. 
  2. Thank you for updating the ECAB reference and also clarifying some of my points in my earlier response. I still have 3 persistent comments that will help with overall clarity for the reader. 1. please note - as you did in your response to me - that this is not a validated tool 2. please note in the methods that the ECAB was administered to only patients on benzodiazepines. 3. related to 2, I noticed that you mentioned that "the ECAB was scored for 176 of the 185 patients taking benzodiazepines." However, I am only counting 184 (Lormetazepam [105] + Loprazolam [42] + Estazolam [25] + Nitrazepam [10] + Temazepam [2]). Am I missing something? 

Author Response

January 5th, 2022

Dear Dr Nelson,                                                        

We are extremely grateful for your interest in our manuscript and the reviewers’ very constructive comments. Please find enclosed a revised version of the manuscript entitled: "Sleep medication in older adults: identifying the need for support by a community pharmacist".

In line with the reviewers’ comments, we have improved the presentation and analysis of data and the contents in general. Point-by-point responses to the reviewers are given below.

Reviewer 2 reports:

Overall most of my comments were addressed. There are still a few points that are unclear after your revision.

Comment

I can appreciate that the authors do want want to convolute the message regarding the univariate and multivariate analyses that identify factors that may be associated with a higher likelihood of wanting to deprescribe. However, I still I think you should include these analyses in some capacity.  Perhaps the data can be placed in the appendix as supplemental tables to avoid - as you state - weighing your paper down with stats. Then, summarize the results as you did in your response to me in the results section.

Response

We have added in the “Statistical analysis” part:

Logistic regression analysis was used to identify independent predictors of a wish to withdrawal hypnotic treatment. For each continuous variable, the log-linearity assumption was assessed with a visual inspection of the scatterplot between the empirical logits and the covariate, and was tested by comparing a model with the continuous covariate to a model including a quadratic component with an F-test for nested models. The multivariate model was built by including all predictors, using manual backward selection to reduce the model, by maximising the c-statistics. The final model was assessed with Hosmer-Lemeshow goodness of fit test. The two-sided type I error rate was set at alpha=0.05.

And in the “results” part:

The multivariate analysis showed that young patients, a decrease in dosage in the last 6 months, a change in dosage by the patient himself and the patient's management of the treatment were positive factors for the patient to wish to withdrawal treatment (Figure 3).

Figure 3. Multivariate logistic regression analysis to identify factors of a wish to withdrawal hypnotic treatment.

Comment

Also, it would be helpful to revise tables 2 and 3 similar your revised table 1.

Response

We have changed tables 2 and 3.

Table 2: Cause of nocturnal awakening. Data were missing for 9 of the 960 patients.

Patients were not ready to try discontinuing (n=706)

Patients who were ready to try discontinuing  (n=245)

Whole sample

(n = 960)

Nycturia

342 (48%)

137 (56%)

479 (50%)

Stress/anxiety

86 (12%)

34 (14%)

120 (13%)

Pain

68 (10%)

27 (11%)

95 (10%)

Nightmares

40 (6%)

26 (11%)

66 (7%)

Noise outside

44 (6%)

21 (9%)

65 (7%)

Spouse

36 (5%)

21 (9%)

57 (6%)

Snoring

26 (4%)

20 (9%)

46 (5%)

Noise in the house

18 (3%)

14 (6%)

32 (3%)

Respiratory problems, nocturnal cough

21 (3%)

8 (3%)

29 (3%)

Animals/pets

17 (2%)

5 (2%)

22 (2%)

Hot flushes

11 (1%)

10 (4%)

21 (2%)

Gastroesophageal reflux

8 (1%)

13 (5%)

21 (2%)

Table 3Hypnotic medications

Patients were not ready to try discontinuing (n=706)

Patients who were ready to try discontinuing  (n=245)

Whole sample

(n = 960)

Current hypnotic medications [n(%)]1

Zolpidem

354 (50%)

107 (44%)

461 (48%)

Zopiclone

183 (26%)

71 (29%)

254 (27%)

Lormetazepam

72 (10%)

33 (13%)

105 (11%)

Loprazolam

30 (4%)

13 (5%)

43 (4%)

Estrazolam

20 (3%)

5 (2%)

25 (3%)

Hydroxyzine

14 (2%)

7 (3%)

21 (2%)

Doxylamine

14 (2%)

2 (<1%)

16 (2%)

Nitrazepam

7 (1%)

2 (<1%)

9 (1%)

Melatonin

5 (<1%)

1 (<1%)

7 (<1%)

Alimemazine

5 (<1%)

0 (0)

5 (<1%)

Temazepam

0 (0%)

2 (<1)

2 (<1%)

Promethazine

0 (0%)

1 (<1)

1 (<1%)

Dosage in compliance with the SPC* [n(%)]2

543 (77%)

186 (76)

729 (78%)

Duration of treatment (months) [median [interquartile range]]3

108 [48 – 240]

120 [36 - 168]

120 [48 - 180]

Dosage change by the physician in the last 6 months [n(%)]4

79 (11%)

47 (19%)

126 (13%)

Dosage change by the patient at some point [n(%)]5

187 (26%)

123 (50%)

310 (33%)

Takes at least one more dose during the night [n(%)]6

94 (13%)

41 (17%)

135 (14%)

Management of hypnotic medications by the patient him/herself [n(%)]7

619 (88%)

226 (92%)

845 (88%)

ECAB score8 ³ 6

63 (9%)

26 (11%)

89 (51%)

Comment

Thank you for updating the ECAB reference and also clarifying some of my points in my earlier response. I still have 3 persistent comments that will help with overall clarity for the reader. 1. please note - as you did in your response to me - that this is not a validated tool 2. please note in the methods that the ECAB was administered to only patients on benzodiazepines. 3. related to 2, I noticed that you mentioned that "the ECAB was scored for 176 of the 185 patients taking benzodiazepines." However, I am only counting 184 (Lormetazepam [105] + Loprazolam [42] + Estazolam [25] + Nitrazepam [10] + Temazepam [2]). Am I missing something?

Response

The tool has been validated on 4,425 patients on benzodiazepine (Pélissolo, A. et al. Anxiety and depressive disorders in 4 425 long term benzodiazepine users in general practice. L’Encéphale2007, 33, 32–38, doi:10.1016/S0013-7006(07)91556-0.).

The tool is not validated on other classes as we said in our previous response (Z drugs, anticholinergics, melatonin, etc.).

We have added this sentence in the methods section: “The patient’s dependency on benzodiazepines was assessed on the Cognitive Scale for Benzodiazepine Attachment (Echelle Cognitive d'Attachement aux Benzodiazépines, ECAB) and was performed only patients on benzodiazepines”.

We thank you for your vigilance, as it is indeed 184 patients. We have changed the sentence: “The ECAB was scored for 176 of the 184 patientstaking benzodiazepines.”
